# COVID-19 and Nanoscience in the Developing World: Rapid Detection and Remediation in Wastewater

**DOI:** 10.3390/nano11040991

**Published:** 2021-04-12

**Authors:** Muhammad Adeel, Tahir Farooq, Noman Shakoor, Sunny Ahmar, Sajid Fiaz, Jason C. White, Jorge L. Gardea-Torresdey, Freddy Mora-Poblete, Yukui Rui

**Affiliations:** 1Beijing Key Laboratory of Farmland Soil Pollution Prevention and Remediation, College of Resources and Environmental Sciences, China Agricultural University, Beijing 100193, China; Chadeel969@gmail.com (M.A.); lb20203030039@cau.edu.cn (N.S.); ryk@cau.edu.cn (Y.R.); 2Guangdong Provincial Key Laboratory of High Technology for Plant Protection, Guangdong Academy of Agricultural Sciences, Guangzhou 510640, China; tahirfarooq@ippcaas.cn; 3Institute of Biological Sciences, Campus Talca, Universidad de Talca, Talca 3465548, Chile; sunny.ahmar@yahoo.com; 4Department of Plant Breeding and Genetics, The University of Haripur, Haripur 22600, Pakistan; sfiaz@uoh.edu.pk; 5The Connecticut Agricultural Experiment Station, New Haven, CT 06504, USA; Jason.White@ct.gov; 6Department of Chemistry and Biochemistry, The University of Texas at El Paso, El Paso, TX 79968, USA; jgardea@utep.edu

**Keywords:** SARS-Cov-2, wastewater, nanoscience, remediation, epidemic

## Abstract

Given the known presence of SARS-Cov-2 in wastewater, stemming disease spread in global regions where untreated effluent in the environment is common will experience additional pressure. Though development and preliminary trials of a vaccine against SARS-CoV-2 have been launched in several countries, rapid and effective alternative tools for the timely detection and remediation of SARS-CoV-2 in wastewater, especially in the developing countries, is of paramount importance. Here, we propose a promising, non-invasive technique for early prediction and targeted detection of SARS-CoV-2 to prevent current and future outbreaks. Thus, a combination of nanotechnology with wastewater-based epidemiology and artificial intelligence could be deployed for community-level wastewater virus detection and remediation.

## 1. Perspective

As of March, 2021, Europe, the People’s Republic of China, and several other countries are bracing for the second wave of deadly Covid-19, and parts of the United States see deeply concerning spikes in reported cases, with a death toll surpassing 545,022. Globally, infections have risen to 127,877,462 persons, with 2,796,561 deaths reported by the World Health Organization (WHO) [1]. The estimated global mortality rate has surged to 2.20%, with new data from the WHO and others suggesting that airborne transmission may be significant [2]. In addition, recent concerns have emerged that the SARS-CoV-2 virus, the Covid-19’s causal agent, may also be spread through virus-containing fecal material. As reported, there has been accumulating evidence of a fecally-mediated route of transmission of SARS-CoV-2 [3]. This rapidly evolving pandemic urgently demands that the scientific community analyzes and evaluates the potential unknown sources of virus transmission that may contribute to Covid-19’s resurgence, as well as effective approaches to mitigate its spread through the already known pathways [4].

In addition to the confirmed routes of Covid-19 transmission, it is possible that wastewater infrastructure can harbor and facilitate the superspreading of the virus [5]. Recent studies have reported the positive detection of SARS-CoV-2 in human wastewater in China [6], Australia [7], France [8], Italy [9], the Netherlands [10], and the USA [11] (Figure 1). Recent findings from Japan mirror these studies, reporting the positive SARS-CoV2 detection at wastewater treatment plants [12]. In countries such as Canada, Germany, and the United States, researchers are actively testing sewage water for coronavirus presence as part of public health surveillance programs. Details of the recent reports on the detection of SARS-CoV-2 are summarized in Table 1. Importantly, the reports indicate that coronaviruses can remain viable in sewage water for up to 14 days, although this is subject to environmental influence [13]. However, SARS-CoV-2 viability under a range of environmental conditions is poorly understood, and the elucidation of these processes is needed to minimize the negative impacts on human health. It is important to note that in many developing countries, wastewater is released (>95%) into the environment without significant treatment [14]. With the exponential rise in the COVID-19 incidence in 219 countries, territories or areas, there is a significant possibility that wastewater exposure may be contributing to pathogen transmission. SARS-CoV-2 has been reported to remain stable and potentially contagious in water and sewage for a period of days to weeks, and, as such, the fecal contamination of water systems may be a significant route of exposure. These findings further highlight the need for active SARS-CoV-2 surveillance in wastewater, particularly in areas where the release of untreated water is common. The development and establishment of a standard protocol for accurate viral quantification in wastewater will significantly advance efforts for monitoring the presence of Covid-19 in the environment.

Nanoengineering science plays a pivotal role in a range of disciplines (agricultural, environmental and medical) that impact everyday life [15,16]. Nanomaterials’ application to mitigate the biological contamination of water has been a topic of robust research and development for a number of years. Engineered nanomaterials such as carbon and metal-based nanoscale filter membranes have been shown to effectively eradicate pathogens and viruses in water systems, and some very important work targeted at understanding the underlying mechanisms of the critical bio-nano interactions has been completed [16,17].

Recent innovations in nanotechnology have involved several useful nanoscale materials for wastewater treatment, including nanometals, nano adsorbents, photocatalysts, and nanomembranes [25]. The use of nanofibers/particles and composite membranes has been demonstrated to successfully remove viruses, bacteria, protozoans and other contaminants from wastewater [26]. For instance, polysulfone ultrafiltration membranes containing silver nanoparticles (Ag-NPs) have shown efficacy against the virus, namely through MS2 bacteriophage removal [27]. In fact, Ag-NPs are considered as highly promising for water remediation due to their cost-effectiveness and high antimicrobial activity [28]. Similarly, titanium dioxide (TiO_2_) films exhibit photocatalytic properties that have both antibacterial and antiviral properties against Escherichia coli and Herpes simplex virus (HSV-1), respectively [29]. Others have proposed the use of electrospun nanofiber membranes (ENMs) to treat the pathogen-contaminated wastewater. Nanofibers composed of ammonium tetrathiomolybdate (ATTM) and tetraethoxysilane (TEOS) blended with polyacrylonitrile (PAN) have a rough and heterogeneously branched surface that enhances bacteria (Escherichia coli, Staphylococcus aureus and Vibrio cholerae) removal efficiency up to >90% [30]. The choice of ENMs (prepared by advanced electrospinning) offers a versatile, scalable, cost-effective and broad range of NMs. In particular, fibrous, ultrathin NMs with specific surface topology and encapsulating and sensing characteristics are high-performance materials, introducing new horizons in different fields—for example, biosensors, drug delivery and regenerative medicine [31,32]. Therefore, the use of nanoparticles alone or as part of complex composite membranes impregnated with different nanomaterials can remove or degrade a range of contaminants, including viruses. Given the above discussion, it is highly likely that these approaches could be effective against SARS-CoV-2. For example, untreated wastewater harboring SARS-CoV-2 could be treated using inexpensive and widely deployable composite filters/membranes containing a range of possible nanomaterials.

Given the known presence of SARS-CoV-2 in wastewater, it is essential to investigate how this impacts overall human exposure and to consider and implement strategies that can mitigate this pathway of viral spread. For example, the impact of virus-contaminated water on consumable aquatic organisms and the possibility of transmission to humans through the food chain should be evaluated. The issues of accidental/unintentional mixing of drinking water and pathogen-harboring wastewater will occur in developing regions, and an understanding of the dynamics of virus activity in this scenario, as well as means to mitigate spread and infection, are critical. The fate of SARS-CoV-2 in a range of water systems under different conditions should be investigated to answer important questions concerning its retention, biological interactions, viability, spatial distribution and transmission.

The potential use of nano-sensors for early detection and monitoring of SARS-CoV-2 should be explored under conditions specific to each developing region. Moreover, these platforms can be combined with cost-effective wastewater-based epidemiology (WBE) techniques [33,34] for effective surveillance, as well as the prediction of community transmission and possible future outbreaks in the less developed regions. More recently, WBE-based technology has been successfully deployed for near-source tracking (NST), permitting the detection of individually existing or clustered infection cases in the sewage-drain-serving infrastructure. Subsequently, a combination of NST and specific clinical testing is now being used to stop disease outbreaks in several countries, including France, Turkey, Estonia, Singapore, USA, UK and Finland [35]. On the other hand, developing countries and rural/impoverished communities within developed regions may be at greater risk of frequent Covid-19 outbreaks due to inadequate infrastructure for the removal of contaminants from wastewater (Figure 2). Given the lack of appropriate detection and remediation measures in the developing nations, this weakness in global public health is a source of concern not only in the current but also in future pandemics. Moreover, a changing climate and stressed biosphere will most certainly push more of these events forward.

Nanotechnology for wastewater virus remediation in developing regions is an emerging field, and the efficacy of this approach will increase with the advancements in nanoscience and a further understanding of nanomaterials–virus interactions. For example, future nanotechnology-based wearable sensor platforms linked with artificial intelligence or the integration with information technology (IT) may improve our understanding of transmission and infection. The possible integration of these fields can enable a better data acquisition and improve the design of nanoparticles for precise remediation in wastewater. A better control over matter at the nanoscale, to improve the efficiency of the reactivity of nanoparticles and the use of artificial intelligence linked with sensing chips, could also become important tools of public health.

## 2. Conclusions

Our perspective highlights novel SARS-CoV-2 threats to water resources and human health. There is an imperative need to explore and analyze the potential unknown sources of SARS-CoV-2 that might cause a resurgence of the virus in the future. An integrated approach involving nanoscience can facilitate the early detection (non-invasive) [36] and effective remediation of future pandemics (Figure 3). For instance, a recent study describes the development of a gold nanoparticle-based colorimetric system that facilitates the “naked-eye” detection of SARS-CoV-2. This assay specifically targets the N-gene of SARS-CoV-2 and the selective agglomeration of AuNPs against target RNA sequence results in a visually detectable precipitate that subsequently minimizes the detection time to less than 10 minutes. Such diagnostic systems are rapid, reliable and convenient, with the potential for a large-scale/community-level diagnosis [37]. Hence, it is clear that existing knowledge can be used to develop technologies that can be rapidly deployed at a global scale to mitigate this deadly threat. Despite the recent surge in nano-enabled antiviral research, the integration of nanoscience for the detection and management of SARS-CoV-2 (and other pathogens) encompasses significant challenges. Importantly, the aspect of the development and deployment of NPs as diagnostic/antiviral agents demands the consideration and deep understanding of several key points. For example, a proper selection and synthesis of NPs with suitable characteristics, their interactions with other molecules based on physicochemical properties and their behavior in various eco/biosystems are the key determinants for the optimization, planning and implementation of sustainable disease management strategies.

## Figures and Tables

**Figure 1 nanomaterials-11-00991-f001:**
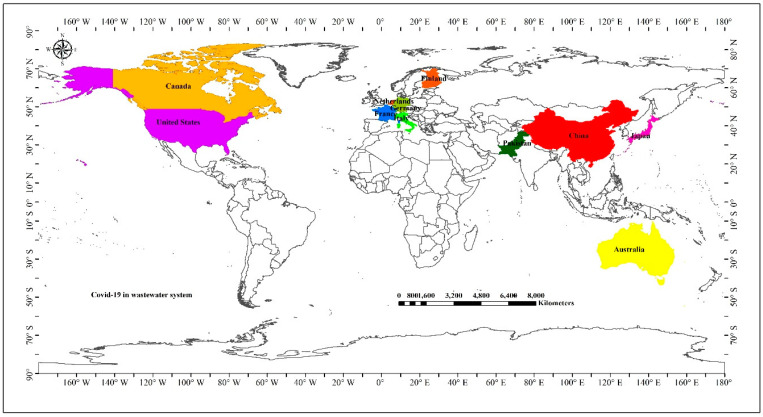
Worldwide detection of SARS-CoV-2 in wastewater until January 2021.

**Figure 2 nanomaterials-11-00991-f002:**
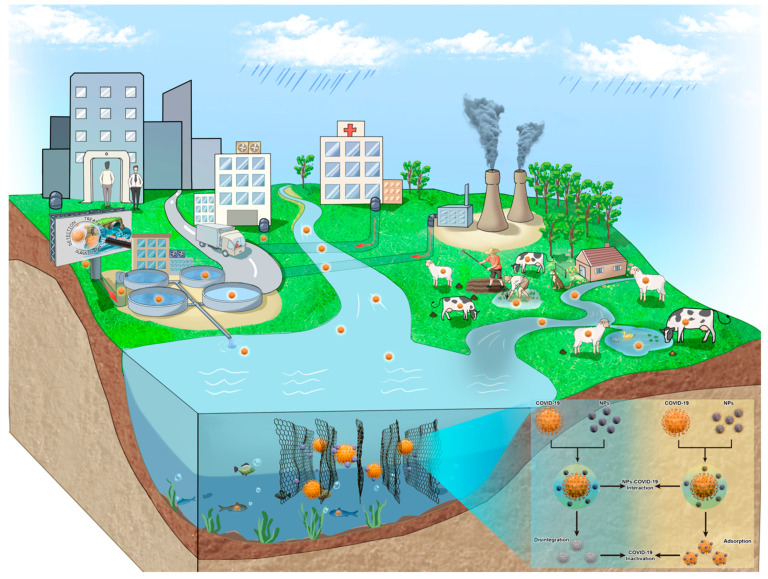
Schematic illustration of primary sources of SARS-CoV-2 in the water system and proposed removal mechanism.

**Figure 3 nanomaterials-11-00991-f003:**
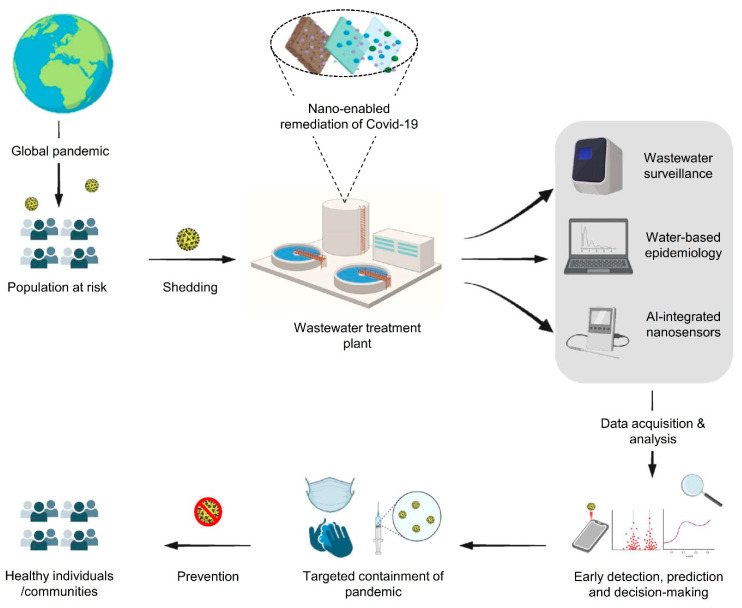
A model representing a combinatory approach for a rapid detection and remediation of SARS-CoV-2 in wastewater.

**Table 1 nanomaterials-11-00991-t001:** Detailed reports on the detection of SARS-CoV-2 RNA in wastewater.

Country	City/County	Specimen Source	Detection Method	Percentage Detection	Concentration (Copies/L)	Reference
Australia	Brisbane, Queensland	Untreated wastewater	RT-qPCR ^c^, sequencging	22% (2/9)	1.2 × 10^2^	[7]
Canada	Ottawa, Gatineau	PCS ^a^	RT-ddPCR, RT-qPCR	90.6–92.7% (*n* = 6)	Not available	[18]
		PGS ^b^		79.2–82.3% (*n* = 5)	Not available	
China	Beijing	Wastewater	RT-qPCR	Not available	Not available	[19]
Finland	Helsinki	Wastewater	RT-qPCR	Pellte (78–89%)	Varied according to assay	[20]
				Supernatent (59–100%)		
France	Paris	Treated wastewater	RT-qPCR	100% (23/23)	>10^6.5^	[8]
		Untreated wastewater	RT-qPCR	75% (6/8)	~10^5^	
Germany	North Rhine-Westphalia	Solid phase wastewater ^d^	RT-qPCR, sequencing	Not available	25 copies/mL	[21]
		Aqueous phase wastewater ^e^			1.8 copies/mL	
Italy	Milan, Rome	Untreated wastewater	RT-qPCR	50% (6/12)	Not available	[9]
Japan	Yamanashi	Secondary-treated wastewater ^f^	RT-qPCR	20%	2.4 × 10^3^	[22]
Netherlands	Amsterdam, Utrecht, Amersfoort, Apeldoorn, Tilburg, Schiphol	Untreated wastewater	RT-qPCR	58% (14/24)	Not available	[10]
Pakistan	38 districts	Untreated wastewater	RT-qPCR	27% (21/78)	Not available	[23]
USA	Massachusett,	Untreated wastewater	RT-qPCR, sequencing	71% (10/14)	>2 × 10^5^	[11]
	Bozeman, Montana	Untreated wastewater	RT-PCR, sequencing	100% (7/7)	>3 × 10^4^	[24]

^a^ Primary clairified sludge, ^b^ Post grit solid, ^c^ Polymerase chain reaction, ^d^ Additional processing by using biological/chemical methods, ^e^ Municipal wastewater treatment plants, ^f^ River water in Yamanashi Prefecture, Japan.

## Data Availability

Data sharing not applicable.

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
