# Peer review of "COVID-19 and Nanoscience in the Developing World: Rapid Detection and Remediation in Wastewater"

_nanomaterials, 2021, doi:10.3390/nano11040991_

Round 1

Reviewer 1 Report

This is an interesting perspective manuscript highlighting the application of nanotechnology in monitoring and in developing strategies in minimizing the risk associated with COVID-19 in wastewater. I will recommend its publication after minor revision.

General comments: The language of the manuscript needs special attention.

Specific comments:  Table 1: Please give in footnote the full-form of "dd" in RT-PCR

 Line 108: Please define the terminology "near-source tracking"

Reviewer 2 Report

  1. In line 39: “WHO and others suggesting that airborne transmission may be significant” Please provide the reference for this statement.
  2. The water cannot transmit COVID-19. The objective/importance of rapid detection and remediation in wastewater need to be clarified.
  3. The authors discussed the issue in a very general manner. This could be adopted for any waterborne pathogens, not specifically COVID-19.
  4. Please provide some references where the nanomaterials have been utilized to detect or treat COVID-19 waste, if available.

Reviewer 3 Report

In this paper, Adeel et al. reviews a promising, non-invasive technique for early prediction and targeted detection of SARS-CoV-2 to prevent current and future outbreaks. This is an interesting study. However, there are a lot of unclear mechanisms on the presence of COVID-19 linked virus and its detection in wastewater. I recommend this paper to reconsider after major revisions. Following are my specific comments;

  • Line 37 – what is 480,464?
  • In the first paragraph, authors should introduce the major transmission routes of SARS-CoV-2 and then they should discuss how it can transmitted via wastewater.
  • Line 41 – ‘agent, may also be spread through virus-containing fecal material’, please specify with examples.
  • Line 44- ‘roaches to mitigate spread through those pathways’, what are those pathways?
  • Line 66-70, needs a reference here. I suggest to cite a recently published paper here; 1016/j.nantod.2020.100962
  • Table 1, when authors refer to sequencing, they should shed light on this methods for readers. Also, I suggest author to write PCR name in full in the first place and they can then use abbreviation.
  • Authors should add a bit of detail on solid/aqueous phase wastewater, secondary treated wastewater and similar terms. Authors have not made such terms clear at all.
  • Line 80 – ‘…..containing silver nanoparticles (Ag-NPs) have shown efficacy at virus removal’. Which virus?
  • Authors have made several statement on viruses but they did not make it clear whether they are referring to enveloped or non-enveloped viruses, as this is significantly important to know before we discuss their removal, detection or treatment. Authors should make it clear.
  • Line 83 ‘….. antibacterial and antiviral properties’. Against which bacterial or viral strains? It is important to know before we discuss the exploitation of such nanosystems for applications in infectious diseases.
  • Line 86 – ‘…. surface that enhances virus removal efficiency up to >90%’. Again which virus? Please add viral, bacterial strains in each study which authors have summarized in this perspective.
  • Line 89 – ‘….and a broad selection of materials’. Which materials and selection on which basis?
  • Line 92 – ‘….. wastewater harboring SARS-CoV-2 could be treated using inexpensive and widely deployable composite filters/membranes containing a range of possible nanomaterials’. Which composites and which nanomaterials and on which basis? Authors should make it clear. I suggest to add a table on this. As such table will be helpful in terms of types and forms nanoparticles, their size, properties, antibacterial, antiviral action and mechanism of action.
  • Line 104 – ‘…. use of nano-sensors for early….’. Can authors please specify the examples of such nanosensors. Authors should cite some other relevant reviews on this subject here. For example, as authors propose non-invasive detection. For non-invasive detection coherent anti-Stokes Raman scattering microscopy can be used. https://doi.org/10.1557/mrc.2020.81
  • Conclusion is too short. I suggest adding limitations and solutions in this section. Authors should provide key insights into the potentials and challenges of the system they are proposing.
  • Line 138 – ‘…. can be rapidly deployed at a global scale to mitigate this deadly threat’. How it can be rapidly deployed. This is a misleading statement. Authors should provide background and strong evidences to such statement.

Round 2

Reviewer 2 Report

Please provide the reference for Figure 1, if you have adopted from another source.

Author Response

Dear Reviewer,

Figure has been drawn by himself and Data was obtained from the publish articles,Which already being discussed and cited properly in the manuscript.

Reviewer 3 Report

I am pleased to recommend the revised manuscript for publication in Nanomaterials.

Author Response

Thank you very much for helps us to improve our MS.